# The Impact of Urban Construction Land Change on Carbon Emissions—A Case Study of Wuhan City

**DOI:** 10.3390/ijerph20020922

**Published:** 2023-01-04

**Authors:** Yuchuan Tan, Yanzhong Liu, Yong Chen, Zuo Zhang, Dan Wu, Hongyi Chen, Yufei Han

**Affiliations:** 1College of Resource and Environmental Engineering, Wuhan University of Science and Technology, Wuhan 430081, China; 2School of Public Administration, Central China Normal University, Wuhan 430077, China

**Keywords:** UCL, carbon emissions, spatial and temporal variation, Wuhan

## Abstract

Urban construction land (UCL) change is a significant cause of changes in urban carbon emissions. However, as the extent of this effect is currently unclear, cities cannot easily formulate reasonable carbon reduction policies in terms of land use. Taking the city of Wuhan, China, as an example, this paper combines data on land use and carbon emissions from 1995 to 2019 and uses spatial analysis, curve estimation, and correlation evaluation to explore the direct and indirect effects of the UCL changes on carbon emissions. The results show that: (1) Between 1995 and 2019, the UCL area in Wuhan increased by 193.44%, and carbon emissions increased by 78.63%; moreover, both changes showed a gradually increasing spatial correlation, and the quantitative relationship could be better fitted with a composite function model; (2) The UCL change had mainly an indirect impact on carbon emissions via factors such as population and energy use intensity per unit of carbon emissions; (3) The maximum value of carbon emissions inside a unit area decreased during the study period, with an average annual decrease of about 2.02%. Therefore, the city of Wuhan can promote the achievement of its carbon emissions reduction targets by improving the existing land use policies, for example, by dividing the city into multiple functional zones.

## 1. Introduction

Large amounts of carbon emissions can lead to global meteorological changes, which can cause serious natural disasters [1,2]. In Climate Change and Land, the IPCC (Intergovernmental Panel on Climate Change) pointed out the interactive relationship between land-use change and climate change and that reasonable land-use management policies can help achieve the carbon emissions reduction targets of the Paris Agreement [3]. Some scholars have demonstrated that land-use change significantly impacts carbon emissions and meteorological changes [4,5,6]. Land-use change affects not only the carbon stock of soil but also carbon emissions from human activities by changing the linkages between socioeconomic and natural systems [7,8,9]. Land-use change is also the second cause of the increase in carbon emissions from fossil energy combustion [10,11,12]. Therefore, it is necessary to clarify the extent of the impact of land-use change on carbon emissions to support the formulation of rational land-use policies and carbon emissions reduction measures.

Current research on the relationship between land-use change and carbon emissions has focused on the following hot topics:

(1) Determining the role of land-use change on carbon emissions. Houghton et al. [13] explored the relationship between land-use change and carbon emissions by taking Asia as the research object; they found that forestry activities and land-use change in South and Southeast Asia released 43.5 Pg of carbon into the atmosphere during the period 1850–1995. By reconstructing Land-Use and Land-Cover Change (LUCC) data, Pacala et al. [14] found that carbon emissions from terrestrial ecosystems in the United States from 1700 to 1945 were about 27 ± 6 Pg. Ge [15] took China as a case study and used the “thin record model” to measure carbon emissions due to land-use change in the previous 300 years. Later studies focused on the effect of small-scale land-use change on carbon emissions. Ren et al. [16] took Dongliao County, China, as the research object and used land-use change data from 1980 to 2018 to study the effect of land-use change on the carbon stock of the ecosystem.

(2) Land-use carbon emissions accounting methods and standards. In 1980, Houghton [17] proposed and refined a thin-notation model based on an annual time-series bookkeeping model with extensive survey and empirical data, which laid the foundations for numerous subsequent studies proposing models to estimate carbon emissions. Ge [15] used a model estimation method to measure the changes in carbon emissions due to land-use change in China over the previous 300 years; Fang et al. [18] studied the forest vegetation carbon pool in China and its spatial and temporal variation by using the resource inventory data of the Senjin system and associated statistical records in China in the past 50 years. Zhao et al. [19] used remote-sensing statistics and Gross Primary Productivity (GPP) data to build a model to estimate carbon emissions changes in the United States. Although these three methods are widely used for carbon emission calculation, due to the complexity and variability of the underlying data, classification system, research methods, and empirical parameters, accounting results can vary greatly for the same research object. Therefore, it is extremely important to employ a reasonable carbon emissions accounting standard [20]. The National Greenhouse Gas Inventory Program (NGGIP), a thematic working group under the IPCC, has established a database of greenhouse gas emission factors, which is regularly updated and provides a basis for carbon emission accounting in various countries and regions [21]. Fang et al. [22], Zheng et al. [23], Lai [21], Zhang et al. [24], and Ye et al. [25] estimated the carbon sinks of various land types in China using agricultural statistics, remote sensing images, ground observation data, and previous research results, and obtained carbon emission factors for forest land, cropland, unused land, watershed, and grassland, which provided a basis for later scholars to use the factor measure to calculate the carbon emissions of different regions of China.

(3) The mechanism of land-use change on carbon emissions. Xia et al. [26] used ecological network analysis to explore the ecological relationship between different land-use changes and proposed a land-carbon correlation rate to describe the impact of land-use changes on carbon balance. Yuan et al. [27] explored the relationship between urbanization and land-use change in three representative models by simulating land use in 13 cities in the Beijing–Tianjin–Hebei urban agglomeration, China and using environmental Kuznets curves; they found that land-use patterns at different levels of urbanization have other effects on carbon emissions. Rounsevell [28] analyzed the impact of land-use change on carbon emissions in the UK, finding that socioeconomic and technological changes may be the most important drivers of land-use change, which in turn determines carbon emissions changes. The above-mentioned studies have initially revealed the role of land use on carbon emissions; however, they mainly focused on analyzing the impact of land-use changes on carbon emissions in provincial areas. As such, they have the following shortcomings: (1) They lack an analysis of the impact of land-use changes on carbon emissions within cities from a spatial perspective and fail to reveal the extent of the influence of land-use changes on carbon emissions in a comprehensive way by establishing quantitative models; (2) They lack an urban-scale exploration of carbon emissions from land use, and the existing carbon emission assessments at the city level are limited to the estimation of energy consumption [29,30,31], which is not helpful for urban land use and carbon reduction development, and does not allow to provide more precise guidance.

In recent years, with the implementation of the national strategy of the “Yangtze River Economic Belt”, the industrialization and urbanization level of the city of Wuhan, which is located in the middle reaches of the Yangtze River, has been rapidly advancing, and the UCL area has been rapidly increasing. The population size, technology development level, and energy use have changed accordingly, affecting urban carbon emissions and posing a remarkable risk to the city’s sustainable development. Therefore, the issue of optimizing the layout of land use and coordinating the relationship between land use and carbon emissions has become urgent for the city of Wuhan. In this study, we quantitatively evaluated the spatial and temporal variation characteristics between the UCL area and carbon emissions in Wuhan and determined the spatial connection between UCL change and carbon emissions change through spatial correlation analysis; then, we quantified the degree of impact of the UCL change on carbon emissions using curve estimation, and analyzed the relationship between carbon emissions influences, identified by Kaya’s constant equation, and changes in UCL using grey correlation; finally, we established the direct and indirect relationship between the change in UCL area and the change in carbon emissions in Wuhan city during the study period. The results of this study can provide suggestions for cities to formulate rational land use policies and promote sustainable urban development.

## 2. Materials and Methods

### 2.1. Materials

#### 2.1.1. Study Area

Wuhan is located between 29°58′–31°22′ N latitude and 113°41′–115°05′ E longitude, at the confluence of the Yangtze River and the Han River. This area is characterized by several lakes and a well-developed water system; the landscape is low and flat in the central part, hilly in the northern and southern parts, and low mountainous in the north (Figure 1).

With the implementation of the Outline of the Yangtze River Economic Belt Development Plan in September 2016, Wuhan has become the main city for the development of the Yangtze River Economic Belt. Therefore, its ability to achieve efficient and low-carbon land use in rapid development is of tremendous importance for the future construction of an ecologically prioritized, green, coordinated, and sustainable society; moreover, Wuhan will also play a leading position in the low-carbon improvement of neighboring cities.

#### 2.1.2. Data Collection

The main data sources employed in this study are shown in Table 1.

### 2.2. Methods

#### 2.2.1. Calculation of Carbon Emissions

The coefficient measurement method was used to calculate carbon emissions. This method is easy to implement, has robust convincing power, and is widely used to calculate carbon emissions [26,27]. The calculation formula employed is as follows:(1)$$C=\sum_{i=1}^{6}C_{i}=\sum_{i=1}^{6}A_{i}\times S_{i},$$
where *C* represents the total carbon emissions;$\;C_{i}$ represents the carbon emissions for each land category; $A_{i}$ represents the carbon emission factor for each land category; and $S_{i}$ represents the utilized area for each land category. *i* was assigned a value from 1 to 6 to indicate Cropland, Water, Forestland, Grassland, Unused land, and UCL, respectively.

The human activities in UCL vary from city to city; hence, using a fixed carbon emission factor was impossible. However, as UCL mainly hosts social production activities, in this study, the method of Xiao et al. [33] was adopted, which expresses UCL carbon emissions (*C*_6_) in terms of carbon emissions generated through energy consumption as follows:(2)$$C_{6}=\sum B_{t}\times Q_{t},$$
where *B_t_* represents the carbon emission factor of each energy source; and *Q_t_* represents the quantity of use of each energy source.

The carbon emission coefficients of Forestland, Unused land, Water, Grassland, and Cropland in Wuhan and the carbon emission coefficients of major energy sources are shown in Table 2.

#### 2.2.2. Land-Use Changes Dynamic Attitude

Land use dynamic attitude represents the magnitude of changes in the way various land-use categories are utilized over a certain period and can be used to quantitatively measure the magnitude of land-use change [34]. In this study, we used a single land-use dynamic attitude to assess the change in land-use types in Wuhan, as follows:(3)$$LC_{i}=\frac{U_{b}-U_{a}}{U_{a}}\times \frac{1}{T}\times 100\%,$$
where $LC_{i}$ represents the dynamic attitude of using single land areas; *U_a_* and *U_b_* represent the area of single land use sorts at the beginning and the end of the study period, respectively, and *T* represents the temporal interval.

#### 2.2.3. Spatial Correlation Analysis

In this study, the spatial autocorrelation analysis was used to explore the spatial and temporal characteristics of the impact of the UCL change on carbon emissions in Wuhan. This type of analysis is articulated into global and local spatial autocorrelation analysis [35].

Global spatial autocorrelation analysis can be used to determine the average degree of correlation and spatial distribution between attributes of a region and to reflect the similarity between each unit and the neighboring units in the entire study area. The Moran’s *I* index is often used to measure global spatial correlation. Its value is in the range [−1, 1]; higher values indicate a stronger correlation between the overall attributes of a region. The *Moran’s I* index was calculated as follows [36]:(4)$$I_{pq}^{i}=\frac{Y_{p}^{i}-{\bar{Y}}_{p}}{T_{p}^{2}}\cdot \sum_{j=1}^{n}W_{ij}\cdot \frac{Y_{q}^{i}-{\bar{Y}}_{q}}{T_{q}^{2}},$$
where $I_{pq}^{i}$ denotes the spatial correlation between the *p*-attribute and the *q*-attribute of the *i*-th spatial unit; $Y_{p}^{i}$ denotes the value of the *p* attribute on the *i*-th spatial unit; ${\bar{Y}}_{p}$ denotes the mean of all spatial unit *p* attributes in the study area; $T_{p}^{2}$ denotes the variance of all spatial unit *p* attributes in the study area; $Y_{q}^{i}$ denotes the value of attribute *q* on the *i*-th spatial unit; ${\bar{Y}}_{q}$ denotes the mean of all spatial unit *q* attributes in the study area; $T_{q}^{2}$ denotes the variance of all spatial unit *q* attributes in the study area; $W_{ij}$ denotes the weight matrix based on the row criterion; and *n* denotes the number of spatial units.

Local spatial autocorrelation analysis can determine the possible spatial correlation patterns and local spatial distribution characteristics of different spatial locations. The autocorrelation of the local space can be analyzed by employing the local Moran’s *I* index. A Local Indicators Spatial Autocorrelation (LISA) clustering map was drawn based on the *Z*-test (*p* < 0.05). Five types of clusters were identified: H-H clusters, where the attribute values of the observed area and its surrounding areas are high, and other clusters are similar; L-L clusters; H-L clusters; L-H clusters; and NS clusters. The formula for the local spatial Moran’s *I* index was as follows [36]:(5)$$I_{i}=\frac{Y_{i}-\bar{Y}}{T_{i}^{2}}\cdot \sum_{i=1,j\neq 1}^{n}W_{ij}\cdot \left(Y_{i}-\bar{Y}\right),$$
where $Y_{i}^{}$ denotes the attribute value of the *i*-th spatial unit; ${\bar{Y}}_{}$ denotes the mean of the attribute values of all spatial units in the study area; $T_{}^{2}$ denotes the variance of the attribute values of all spatial units in the study area; $W_{ij}$ denotes the weight matrix based on the row criterion; and *n* denotes the number of spatial units.

#### 2.2.4. Curve Estimation

To further quantify the influence of the UCL change on carbon emissions, scatter plots were drawn considering the UCL area as the independent variable and carbon emissions as the dependent variable. A suitable mathematical model was selected for curve estimation based on the scatter plot, and the best-fit equation was determined using the significance test. In this study, we use Linear Model (Equation (6)), Logarithmic Model (Equation (7)), Quadratic Model (Equation (8)), Three times Model (Equation (9)), Composite Model (Equation (10)), Power Model (Equation (11)). These formulas employed were as follows:(6)$$y=\beta_{0}+\beta_{1}x$$
(7)$$y=\beta_{0}+\beta_{1}\ln x$$
(8)$$y=\beta_{0}+\beta_{1}x+\beta_{2}x^{2}$$
(9)$$y=\beta_{0}+\beta_{1}x+\beta_{2}x^{2}+\beta_{3}x^{3}$$
(10)$$y=\beta_{0}*\beta_{1}{}^{x}$$
(11)$$\;y=\beta_{0}x^{\beta_{1}}$$

The goodness of match of the regression model was determined by using the equation fit coefficient *R*^2^, whereby the closer the *R*^2^ to 1, the better the model fits the change in carbon emissions and the change in UCL. Moreover, the significance test was employed to determine the validity of the regression equation, whereby the smaller the significance coefficient *p*, the more valid the equation is in responding to the effect of the UCL change on carbon emissions. Generally speaking, this equation is considered extremely valid when *p* < 0.05 [37].

#### 2.2.5. Indirect Impacts of UCL Changes on Carbon Emissions

While the direct impact of UCL changes on carbon emissions was determined by assessing the qualitative and quantitative relationship between these two elements, the indirect impact was determined by assessing the relationship between UCL changes and the factors influencing carbon emissions. The carbon emissions impact factors were then determined by employing Kaya’s constant equation.

Kaya’s constant equation was first proposed by the Japanese scholar Yoichi Kaya [38]. This equation links the general macro factors, such as society and economy, to carbon emissions, and considers carbon emissions as the result of the combined effect of four factors: GDP per capita; energy consumption per 10,000 Yuan GDP; energy use intensity per unit of carbon emissions; and population. Due to its simple structure and robust explanatory power of the change factors, it is widely recommended by the IPCC to analyze the characteristics of carbon emissions changes and its influencing factors [39,40]. As such, it has been widely recommended by the IPCC to analyze the characteristics of carbon emissions changes and their influencing factors. The formula employed is as follows:(12)$$C\propto \frac{C}{E}*\frac{E}{GDP}*\frac{GDP}{P}*P,$$
where *C* is carbon emission; *E* is energy use; *p* is population number; $\frac{C}{E}\;$denotes energy use intensity per unit of carbon emission; $\frac{E}{GDP}$ denotes energy use per 10,000 Yuan; $\frac{GDP}{P}$ denotes GDP per capita.

The gray correlation analysis was used to explore the relationship between UCL changes and factors influencing carbon emissions. The gray correlation analysis can measure the connection between two elements and is suitable for small samples and in cases of poor information and uncertainty. The greater the gray correlation, the closer the relationship between the two elements [41]. The formula employed is as follows:(13)$$\gamma_{0}\left(i\right)=\frac{1}{m}\sum_{k=1}^{m}\xi_{i}\left(k\right),$$
where $\xi_{i}\left(k\right)=\frac{\underset{i}{\min}\underset{k}{\min}\left|x_{0}\left(k\right)-x_{i}\left(k\right)\right|+\underset{i}{\rho \max}\underset{k}{\max}\left|x_{0}\left(k\right)-x_{i}\left(k\right)\right|}{\left|x_{0}\left(k\right)-x_{i}\left(k\right)\right|+\rho \underset{i}{\max}\underset{k}{\max}\left|x_{0}\left(k\right)-x_{i}\left(k\right)\right|}$; $\gamma_{0}\left(i\right)$ and $\xi_{ij}$ represent the gray correlation coefficient and the correlation degree corresponding to $x_{i}\left(k\right)$ and the reference sequence $x_{0}\left(k\right)$, respectively; ρ represents the resolution, and is generally assigned a value of ρ=0.5; and $x_{0}\left(k\right)$ and $x_{i}\left(k\right)$ represent the reference series and the *k*-th term of the *i*-th variable, respectively.

## 3. Results

### 3.1. Analysis of Urban Land-Use Change and Carbon Emissions Change

#### 3.1.1. Urban Land-Use Change Analysis

The results of the spatial analysis of land use in Wuhan from 1995–2019 are shown in Figure 2. UCL was mainly located in the central urban area of Wuhan, while in other areas, it showed a concentrated distribution with multiple land-use centers, as well as a star-shaped spreading pattern from the center to the surrounding area. The total area of UCL increased from 391.55 km^2^ in 1995 to 1148.96 km^2^ in 2019, with an average annual increase of 2.79%. Furthermore, the Cropland area decreased from 6382.86 km^2^ in 1995 to 5573.21 km^2^ in 2019, with an average annual decrease of 33.74 km^2^. The water area is located in the central and southern sections of the city; its extension decreased from 1225.62 km^2^ in 1995 to 1194.268 km^2^ in 2019, following a fluctuating downward trend of increase–decrease–increase–decrease. Forestland is mainly located in the northwestern region and in the northeastern fringe area, while the other land types are scattered in the central and southern regions, with the total area following a trend of first decreasing and then increasing.

In the process of UCL changes, the annual growth showed a fluctuating change of decrease-increase-decrease; the period with the highest growth was 2005–2010, with an average annual growth of 43.06 km^2^, while that with the lowest growth was 2000–2005, with an average annual increase of 23.81 km^2^. The period with the greatest change in land-use dynamics was 1995–2000, reaching as high as 6.91%, while the lowest change was 2015–2019, with 2.88% (Figure 3).

During the study period, the new UCL occupied the largest amount of Cropland, i.e., about 713.97 km^2^, accounting for about 94.40% of the new UCL, followed by water with 34.28 km^2^, accounting for about 4.53% of the new UCL.

Looking at different study periods, the proportion of new UCL originating from Cropland varied widely, with the highest proportion in 2010–2015, accounting for 100.46% of new UCL, and the lowest in 1995–2000, at 90.24%. In addition, the period with the largest proportion of the water conversion of new UCL occurred in 2000–2005, at 8.84%. The proportion of the sum of other land types to the new UCL fluctuated, with the highest proportion of 1.53% calculated for 1995–2000. The proportion of the transformed part of each land type to the new UCL continued to change due to the scattered distribution of multiple land types within Wuhan and the principle of proximity in the expansion of the UCL (Table 3).

#### 3.1.2. Carbon Emissions Analysis

In general, Wuhan’s carbon emissions showed an upward trend, rising from 21,187,800 t in 1995 to 38,972,400 t in 2019, corresponding to an increase of 78.63% in 24 years, with an average annual increase rate of 2.45%. More in detail, carbon emissions from energy consumption on urban construction sites, which is the main source of carbon emissions in Wuhan, increased from 21,594,600 t in 1995 to 38,793,700 t in 2019, with an average annual increase rate of 2.47%. The carbon emissions coefficient of the UCL, determined as the ratio of total carbon emissions to land area, decreased continuously throughout the study period, with an average annual decrease of 2.02%. The energy consumption per 10,000 Yuan GDP in Wuhan also decreased continuously, with an average annual decrease of 10.99%; the largest decline of 64.20% was measured from 2005 to 2010, with an average annual decrease of 18.57% during that period. The decreasing trend of the energy use intensity per unit of carbon emissions, on the other hand, was confirmed, with an average annual decline of 1.42%; the highest decline was measured from 2000–2005, with an average annual decline of 2.67%, while a brief upward trend occurred from 2015–2019. However, this increase was not significant, equaling 1.39% (Table 4).

During the study period, carbon emissions from the direct consumption of fossil energy have always remained above 50% of total carbon emissions (Figure 4), and the consumption of fossil energy such as coal, washed coal, coke, and crude oil has always remained above 75% of the total energy consumption. This is consistent with the fact that China is expected to continue using fossil energy in the future [42].

### 3.2. Spatial Correlation Analysis

ArcGIS 10.2 was used to draw a 1 km × 1 km grid map, where the city of Wuhan was divided into several spatial units according to their spatial locations, which were graded using the natural breakpoint method to obtain the distribution of carbon emissions within a unit area of Wuhan city during the study period. UCL area in the spatial cells was used as the spatial indicator of the UCL, and GeoDa was used to conduct a bivariate spatial analysis of urban land use and carbon emissions.

#### 3.2.1. Urban Land-Use Carbon Emissions

During the study period, the average carbon emissions of each Wuhan area showed an increasing trend. The number of low carbon emissions areas decreased, and the number of other types of areas increased. Moreover, the upper limit for classifying each level of carbon emissions within a unit area also decreased (Table 5).

The analysis of the spatial distribution of carbon emissions per unit area in Wuhan city indicated a gradual decrease from the central to the surrounding areas; more in detail, the overall distribution from north to south and east to west was low-medium-high-medium-low (Figure 5). The high-carbon emissions areas were mainly located in the city center, gradually expanding to the surrounding areas from a scattered to a more integrated distribution, and the area gradually increased. The carbon emissions in the central-northern areas and local northeastern areas gradually increased, while those in the southeastern and southwestern areas generally did not record a considerable change. It is noteworthy that some areas with high carbon emissions were found to have low carbon emissions, and their distribution corresponded to the distribution of water.

#### 3.2.2. Global Autocorrelation

The values of Moran’s I index between UCL and carbon emissions during the study period were all greater than 0 and continued to increase, passing the significance test (Table 6). This indicates an important and increasingly positive correlation between the area of urban land-use change and carbon emissions change.

#### 3.2.3. Local Autocorrelation

Based on the results of the local autocorrelation analysis of carbon emissions and UCL in Wuhan from 1995–2015 (Table 7), it clearly emerged that the overall spatial clustering relationship between carbon emissions and UCL varied greatly over time (Figure 6).

In 1995, the spatial clusters were mainly L-L clusters, and were scattered in various areas of Wuhan; H-H clusters were mainly located in the central part of Wuhan and accounted for a small share of the whole area; in parallel, a large number of L-H clusters were distributed in-between H-H clusters, showing a linear arrangement; finally, H-L clusters were located in the inner part of the city, adjacent to L-L clusters. In 2000, the spatial clustering changed compared to 1995. These changes mainly include the increase in the number of H-H clustering areas, the gradual concentration of L-L cluster areas, and the development trend to the periphery. More in detail, the number of L-H cluster areas decreased; these were mainly located in-between H-H cluster areas, and the distribution pattern did not change much. The number of H-L cluster areas decreased; these were mainly located in the central and southern parts of the city. In 2005, some L-L cluster areas transformed into H-H cluster areas, further increasing their extension.

Moreover, the number of L-L cluster areas continued to enlarge in the peripheral areas of Wuhan; the number of L-H cluster areas increased, and was mainly distributed around the H-H cluster areas, and the number of H-L cluster areas was the same as in 2000. In 2010 and 2015, the number of H-H cluster areas increased continuously; these were mostly located in the central area of Wuhan. In 2010 and 2015, the number of H-H clusters continued to increase; these were located in the central region of Wuhan, with a small number located in the central-northern and central-southern areas. The number of L-L clusters continued to increase, gradually concentrating in the urban periphery, while the area of L-H clusters continued to increase, and the H-L clusters gradually disappeared. In 2019, the areas of concentration of the H-H clusters continued to expand outward, and their number increased compared to 2015. Moreover, the number of L-H clusters decreased by one compared to 2015 and were mainly located near the H-H cluster regions. Finally, the number of L-L cluster regions further increased, and the overall development trend changed less compared to the past.

### 3.3. Impact of UCL Changes on Carbon Emissions

Using UCL area and carbon emission data of Wuhan city for the period 1995–2015, a scatter plot was drawn to represent the changes in UCL and carbon emissions, considering UCL area as the independent variable (x) and carbon emissions as the dependent variable (y), as shown in Figure 7. The linear function model, the quadratic function model, the cubic function model, the composite function model, and the power function model were employed for curve estimation fitting. The obtained fitting results of each function model are shown in Table 8.

As shown in Table 8, although the fit of the function models used in the study (R^2^) was greater than 0.9, the fit of the quadratic function (Equation (14)) and the composite function model (Equation (15)) was higher (R^2^ > 0.99, *p* < 0.01). The following model was thus established:(14)$$y=1815.08512+0.4047x+0.0013x^{2}$$
(15)$$y=1570.5844*1.0008^{x}$$

These two models were used as alternative models, and UCL area and carbon emissions data of Wuhan in 2019 were used as test data to evaluate the prediction accuracy of these two models using the error between the predicted and the true values of the two models and to determine the final curve estimation results. The results obtained by substituting the test data into the alternative model, are presented in Table 9.

The prediction error of the composite model was found to be low; therefore, the composite function model (Equation (15)) was used as the final curve estimation model.

## 4. Discussion

### 4.1. Impact of Spatial and Temporal Changes in UCL on Carbon Emissions

By investigating the land-use scenarios and carbon emissions in Wuhan from 1995–2019, it was found that the UCL area in Wuhan increased by 757.41 km^2^ and carbon emissions rose by 17,154,600 t during the study period. This is the same as the results of Houghton [13], Pacala [14], and Ren et al. [16]. However, unlike this study, Houghton and Pacala et al. derived their results from the analysis of a large study area of terrestrial ecology in Asia and the United States, while Ren explored the effects of land-use change on carbon stocks. In this study, it is worth noting that the largest change and the largest increase in the UCL area occurred in the periods 2005–2010 and 1995–2000, respectively, while both the largest change and the largest increase in carbon emissions occurred in the period 2010–2015. Looking at the spatial distribution of carbon emissions, it was found that carbon emissions from other land types in large low-carbon emission areas increased rapidly when they were converted into UCL. High-carbon emission areas gradually spread from the urban center to the surrounding areas, and gradually connected with high-carbon emission areas in other areas to form a patch, roughly following the same direction as the expansion of UCL. In contrast, the land types in the urban fringe areas mostly played the role of carbon sinks, and their utilization changed less during urban development. Hence, it may be concluded that the carbon emissions in the fringe areas did not change considerably from 1995 to 2019.

At the same time, the present study found that, although overall carbon emissions in Wuhan were increasing, the upper limit of high-carbon emission areas identified by the natural breakpoint method was decreasing; at the same time, total energy consumption per 10,000 Yuan GDP and carbon emission intensity per unit of energy were also found to decrease. These outcomes were mostly due to the continuous enhancement of energy utilization effectiveness and the partial elimination of energy-dependent industries in the process of industrial upgrading thanks to continuous technological advances in Wuhan [43]. This indicates that reasonable carbon emission reduction policies can have an important impact on carbon emissions.

### 4.2. Qualitative and Quantitative Relationships between UCL Changes and Carbon Emissions

The application of the spatial autocorrelation evaluation method allowed us to find a positive spatial correlation between the changes in the UCL area and the changes in carbon emissions in Wuhan, i.e., both changes showed to follow the same spatial development trend. The finding is comparable to those of Li et al. [44], with the difference that the latter used panel records to analyze the effect of land-use change on carbon emissions in Anhui Province from a spatial perspective. In contrast, the finding that changes in the UCL area and carbon emissions were not synchronized needed to be further investigated by building a quantitative model.

The extension of H-H cluster areas increased from 1995 to 2019 in Wuhan, mainly because of the city’s continuous development, such that the other land types around UCL continuously transformed into UCL. The extension of the L-H cluster area changed. The reason mainly lies in that the early development of Wuhan city relied on the convenience of water resources conditions and that the core urban area was built by the river. In that period, Wuhan city vigorously developed tourism and heavy industry, and human activities in water increased together with carbon emissions. After 2010, the development strategy of Wuhan changed; the city began to pay attention to ecological protection, and the carbon emissions in water decreased continuously, determining the spatial clustering of some areas in the form of L-H/H-H/L-H clusters. In the first part of the study period, the L-L cluster areas were scattered between the urban fringe and the H-H cluster areas and then gradually concentrated in the urban fringe. Although urban fringe areas were less affected by land-use changes, the overall average carbon emissions in these areas increased. This is mainly due to the development of the central part of the city and the promotion of the synergistic development of fringe areas, accompanied by an increase in economic activities, which in turn resulted in an increase in carbon emissions. The H-L cluster areas were not found to have high carbon emissions per unit area; this occurred mainly because these areas were composed mostly of Cropland and a small portion of UCL. As Cropland is also a source of carbon emissions, carbon emissions reduction policies for Cropland should also be considered in future development [45,46].

The results of the developed complex function model showed an overall positive relationship between the UCL area and carbon emissions. For every 1 km^2^ expansion of the UCL area, carbon emission increased to reach about 1.001 times the level before expansion. This quantitative relationship proves that the increase of the UCL area increased the generation of carbon emissions.

### 4.3. Relationship between UCL Change and Carbon Emissions Influencing Factors

Using *Kaya*’s constant equation, carbon emissions were decomposed into four factors: energy consumption per 10,000 Yuan GDP; energy use intensity per unit of carbon emissions; GDP per capita; and population. Then, a gray correlation analysis was conducted between these four factors and the UCL area (Table 10). The results of this analysis showed that the three factors of population, energy use intensity per unit of carbon emissions, and energy consumption per 10,000 Yuan GDP were strongly correlated with the UCL area; this indicates that the changes in the UCL area had a strong interaction with these factors and, thus, affected carbon emissions. This is similar to the findings of Yuan [27] and Rounsevell M. et al. [28]: socioeconomic, technological and other elements are considered to play an important role in the process of land-use change affecting carbon emission changes. However, unlike this study, Yuan investigated the mechanism of land-use change on carbon emission from the perspective of analyzing different urbanization levels of cities, in which more socioeconomic and technological indicators are included in the indicators of urbanization level; Rounsevell M investigated the mechanism of land-use change on carbon emission from a macro perspective, taking the UK as an example.

In the future, the relationship between the UCL area and these factors should be coordinated to achieve carbon emissions reduction. The correlation between GDP per capita and the UCL area was poor. This indicates that the changes in the UCL area did not considerably influence the changes in carbon emissions through the interaction with GDP per capita; on the other hand, it also indicates that the increase in the UCL area did not necessarily improve GDP per capita, and urban development should be separated from “blind expansion”.

### 4.4. Study Shortcomings and Future Research

The shortcomings of this study may be summarized as follows:

(1) In this study, the carbon emission coefficients of various land types in Wuhan were calculated by summarizing those derived from previous studies. However, the latter may vary according to the natural vegetation conditions, ground cover, and energy intensity of each place, which may affect the accuracy of the final results.

(2) This study only focused on the impact of spatial and temporal changes in the UCL area on carbon emissions. However, the mechanism of the effects of the UCL changes on carbon emissions is complex. It includes several factors, such as population size and economic development level, which are more or less related to the UCL policy [47]. In the future, we should explore the interaction between these factors and the changes in urban land use and assess how these elements affect carbon emissions through UCL changes from a spatial perspective.

(3) In this study, only the effect of the UCL changes on carbon emissions was analyzed, as this is the major factor affecting land-use change. The assessment of the influence of the UCL changes of other land types on carbon emissions was ignored, which affected the comprehensiveness of the study results.

## 5. Conclusions

Based on previous studies, this study firstly quantifies the characteristics of urban land-use changes in Wuhan city and measures the changes in carbon emissions based on them; after that, using spatial autocorrelation analysis and curve estimation, Kaya’s constant equation and gray correlation analysis, the relationship between spatial and temporal changes of the UCL on carbon emissions is explored from a spatial perspective; finally, the direct and indirect effects of the UCL changes on carbon emissions are determined. The results of the study are as follows: (1) In 2019, the UCL area and carbon emissions in Wuhan were about 2.93 times and 1.79 times those in 1995. The expansion of the UCL area showed to follow a star-shaped spreading from the central area to the surrounding areas, and the areas of carbon emissions increase within the unit area showed an outward expansion in all directions. The spatial distribution and development direction of the areas of carbon emissions increase within a unit area and of the UCL change areas were roughly the same, and were found to have a positive spatial correlation that was increasing year by year. The fitting effect of the composite model on the relationship between UCL area changes and carbon emissions changes in Wuhan was more scientific and rational than other curve estimation models. The proposed model allowed us to find that the growth of the UCL entailed an increase in carbon emissions of about 1.001 times those before the expansion for every 1 km^2^ of the UCL area.

(2) The correlation between UCL area and population, energy use intensity per unit of carbon emission, energy consumption per 10,000 Yuan GDP, and GDP per capita gradually decreased during the study period. More in detail, the correlation between population and energy use intensity per unit of carbon emissions was greater than 0.9, indicating that the UCL area changes will indirectly impact urban carbon emissions by affecting population and energy use intensity per unit of carbon emissions.

(3) The maximum value of carbon emissions within a unit area decreased during the study period, such that the value in 1995 was about 1.63 times that in 2019. This indicates that reasonable policies will positively affect the reduction of carbon emissions, and reasonable land-use policies will promote the achievement of carbon emissions reduction goals in Wuhan on an existing basis.

To achieve the objective of decreasing carbon emissions and promoting sustainable social development, this study suggests adopting the following measures. Firstly, suitable functional areas, such as economic development areas and carbon sink areas, should be established based on the actual situation of each district, avoiding encouraging economic growth and reducing human production activities in the carbon sink areas, as well as strengthening the construction of “satellite cities”. Secondly, we should change our thinking on development, promote technological innovation, optimize and upgrade the existing UCL, improve the resource allocation rate, and promote the optimization and upgrading of existing industries and their development towards low carbonization. Finally, we should make reasonable use of the stock of the UCL, improve land-use conservation, slow down the expansion of the UCL, and give priority to the encroachment of land with weak carbon sink capacity in exchange for the protection of land with strong carbon sink capacity when expanding UCL.

## Figures and Tables

**Figure 1 ijerph-20-00922-f001:**
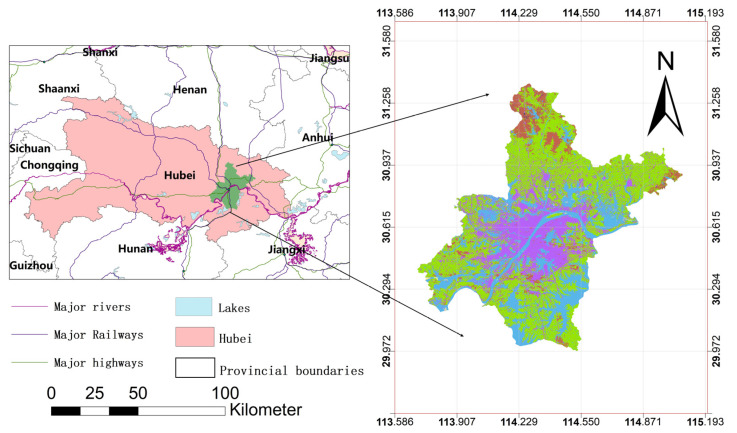
Wuhan city location map. (**a**) Location of Wuhan; (**b**) longitude and latitude of Wuhan.

**Figure 2 ijerph-20-00922-f002:**
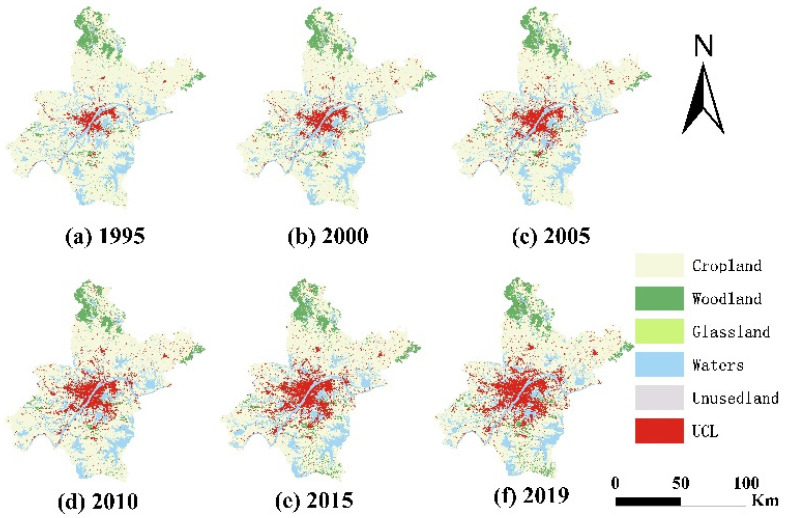
Land use in Wuhan, 1995–2019. (Take six points in time from 1995–2019: 1995, 2000, 2005, 2010, 2015, 2019).

**Figure 3 ijerph-20-00922-f003:**
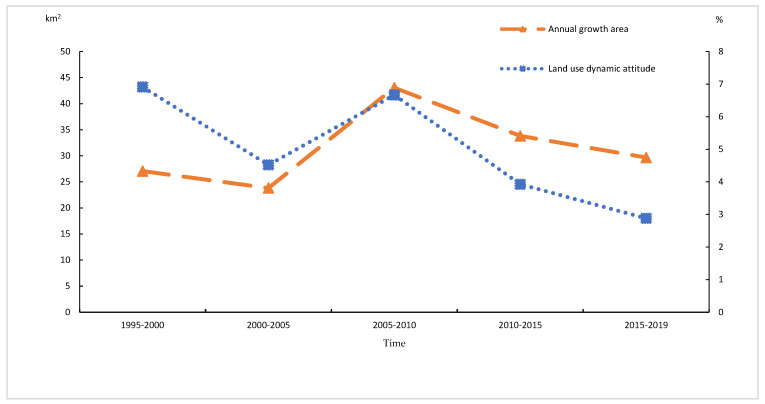
Area of the UCL change and UCL dynamic attitude in Wuhan, 1995–2019 (Divide the years 1995–2019 into five time periods: 1995–2000, 2000–2005, 2005–2010, 2010–2015, and 2015–2019).

**Figure 4 ijerph-20-00922-f004:**
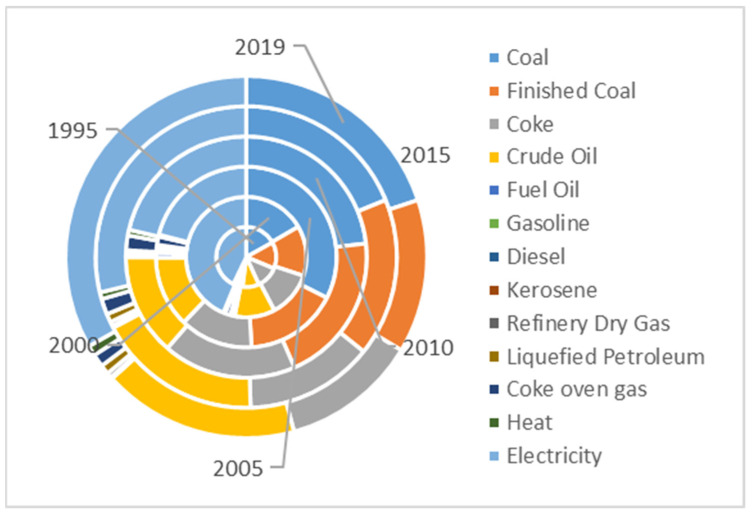
Energy consumption ratio in Wuhan, 1995–2019.

**Figure 5 ijerph-20-00922-f005:**
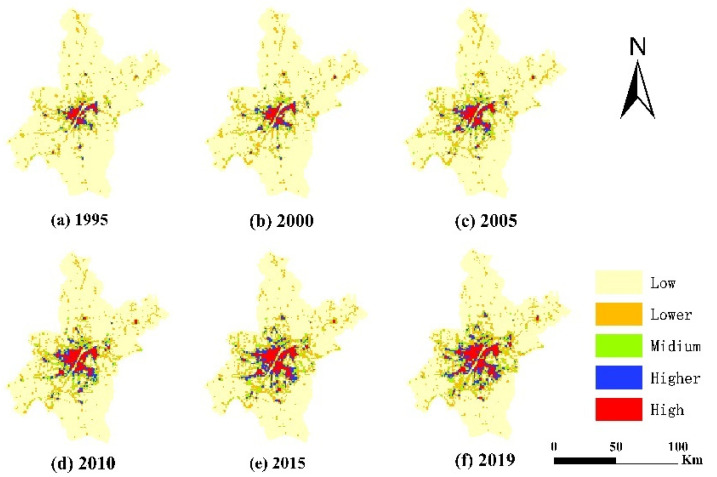
Spatial distribution of carbon emissions within a unit area in Wuhan, 1995–2019.

**Figure 6 ijerph-20-00922-f006:**
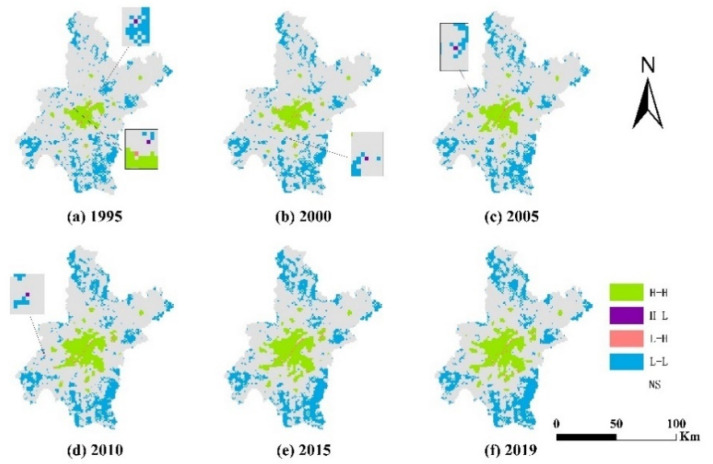
Spatial distribution of spatially localized autocorrelation results between carbon emissions and UCL area in Wuhan, 1995–2019.

**Figure 7 ijerph-20-00922-f007:**
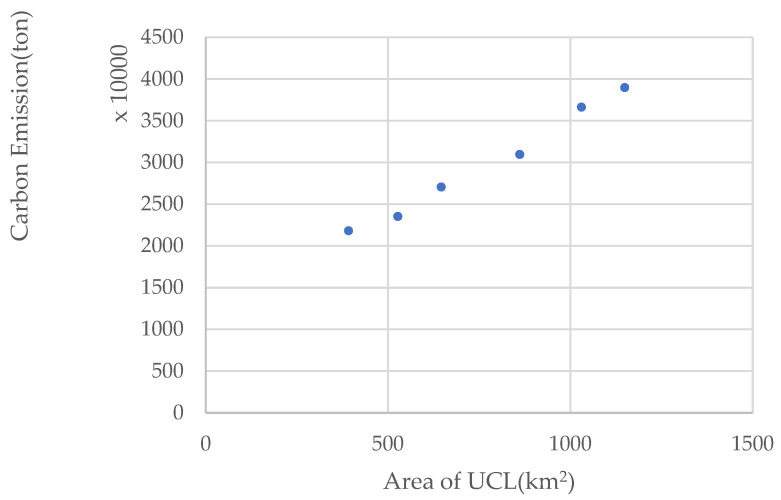
Changes in UCL and carbon emissions, 1995–2019.

**Table 1 ijerph-20-00922-t001:** Data Source.

Name of Data	Source of Data
The land-use data	European Space Agency (ESA) (https://viewer.esa-worldcover.org/worldcover/ (accessed on 9 November 2022))
Socioeconomic and energy consumption data	The Wuhan Statistical Yearbook (http://tjj.wuhan.gov.cn/tjfw/tjnj/ (accessed on 9 November 2022))
Carbon Emission factors of energy source	The 2006 IPCC Guidelines for National Greenhouse Gas Inventories. (https://www.ipcc-nggip.iges.or.jp/public/2006gl/index.html (accessed on accessed on 9 November 2022)) [32]
Carbon emission factors of land types	Research achievements of Li [21], Fang et al. [22], Zheng [23], Zhang et al. [24], Ye et al. [25].

**Table 2 ijerph-20-00922-t002:** Carbon emission coefficients for each type of land use and each major energy source.

Land Use Type	Carbon Emission Factor (t/km^2^)	Energy	Carbon Emission Factor (t/tce)	Energy	Carbon Emission Factor (t/tce)
Cropland	49.7 [23]	Coal	0.7559 [32]	Kerosene	0.5714 [32]
Forestland	−64.4 [22]	FC	0.7559 [32]	RDG	0.4602 [32]
Grassland	−2.4 [25]	Coke	0.8559 [32]	LP	0.5042 [32]
Water	−46 [24]	Crude Oil	0.5857 [32]	COG	0.3548 [32]
Unused land	−0.5 [21]	Fuel Oil	0.6185 [32]	Heat	0.26 [32]
UCL	-	Gasoline	0.5538 [32]	Electricity	2.5255 [32]
		Diesel	0.5921 [32]	BFG	0.3548 [32]

Where the carbon emission factor of the land class is less than zero, it means that the land class is a carbon sink class and has carbon absorption capacity; FC—Finished coal; RDG—Refinery Dry Gas; LP—Liquefied Petroleum; COG—Coke oven gas; BFG—Blast furnace gas.

**Table 3 ijerph-20-00922-t003:** Sources of new UCL in Wuhan, 1995–2019.

	Cropland	Water	Forest Land	Grassland	Wasteland	Totally
1995–2000	122.10	11.12	1.02	0.87	0.18	135.19
2000–2005	107.45	10.52	0.63	0.22	0.15	118.97
2005–2010	200.74	11.57	0.76	1.37	0.24	214.68
2010–2015	169.70	−2.70	0.5	1.38	0.04	168.92
2015–2019	114.08	3.77	0.18	0.48	0.06	118.57
1995–2019	713.97	34.28	3.09	4.32	0.67	756.33

Unit: km^2^.

**Table 4 ijerph-20-00922-t004:** Wuhan Carbon Emissions-Related Indices, 1995–2019.

	Total Energy Consumption Per 10,000 Yuan GDP(t of Standard Coal)	Energy Use Intensity Per Unit of Carbon Emissions	Total Carbon Emissions (10^4^ t)	The Carbon Emission Factor of UCL (10^4^ t/km^2^)
1995	2.78	1.03	2181.78	5.515095
2000	1.43	0.95	2352.93	4.424664
2005	0.81	0.83	2705.73	4.155276
2010	0.29	0.76	3096.16	3.571424
2015	0.19	0.72	3662.59	3.537729
2019	0.17	0.73	3897.24	3.376421

**Table 5 ijerph-20-00922-t005:** Carbon emissions per unit area in Wuhan, 1995–2019.

		Low	Lower	Medium	Higher	High
1995	Upper (t)	2733.84	10,072.52	22,556.94	38,686.81	55,150.90
	Lower (t)	−64.40	2733.85	10,072.53	22,556.95	38,686.82
	Number	7795	695	238	142	146
2000	Upper (t)	2662.07	9228.44	19,773.40	32,708.65	44,246.60
	Lower (t)	−64.40	2662.08	9228.45	19,773.41	32,708.66
	Number	7491	878	294	166	181
2005	Upper (t)	2445.65	8351.33	18,243.34	30,601.90	41,552.80
	Lower (t)	−64.40	2445.66	8351.34	18,243.35	30,601.91
	Number	7124	1042	389	221	232
2010	Upper (t)	2646.76	8532.18	17,194.60	27,131.87	35,714.20
	Lower (t)	−64.40	2646.77	8532.19	17,194.61	27,131.88
	Number	6938	1005	464	286	324
2015	Upper (t)	2773.28	8634.01	17,167.13	26,952.47	35,377.47
	Lower (t)	−64.40	2773.29	8634.02	17,167.14	26,952.48
	Number	6638	1072	534	373	395
2019	Upper (t)	3024.05	9175.16	17,383.56	26,225.59	33,764.20
	Lower (t)	−64.40	3024.06	9175.17	17,383.57	26,225.60
	Number	6559	1074	561	385	437

**Table 6 ijerph-20-00922-t006:** Spatial correlation between carbon emissions and UCL area in Wuhan City, 1995–2019 and the significance test results.

	1995	2000	2005	2010	2015	2019
Moran’s I index	0.799	0.808	0.817	0.829	0.832	0.833
*p*-value	0.001	0.001	0.001	0.001	0.001	0.001
z-value	108.73	111.48	110.88	111.79	112.26	112.30

**Table 7 ijerph-20-00922-t007:** Results of spatial local autocorrelation analysis between carbon emissions and UCL area in Wuhan, 1995–2019.

	1995	2000	2005	2010	2015	2019
H-H	453	538	661	915	1051	1117
H-L	2	1	1	1	0	0
L-H	22	21	29	36	40	39
L-L	1353	1398	1497	1638	1786	1875
NS	7188	7060	6830	6428	6141	5987

**Table 8 ijerph-20-00922-t008:** Summary of curve estimation models and parameter estimates.

Model	Model Summary					Parameter Estimates			
	R^2^	F	df_1_	df_2_	Sig.	Constants	b_1_	b_2_	b_3_
Linear	0.982	162.844	1	3	0.001	1206.105	2.306		
Logarithmic	0.937	44.455	1	3	0.007	−6921.337	1500.012		
Quadratic	0.992	130.818	2	2	0.008	1815.085	0.405	0.001	
Three times	0.993	49.190	3	1	0.104	1177.880	3.477	−0.003	2.186 × 10^−6^
Composite	0.991	329.357	1	3	0.000	1570.584	1.001		
Power	0.964	80.619	1	3	0.003	87.136	0.533		

Dependent variable: Carbon emissions. Independent variable: UCL area.

**Table 9 ijerph-20-00922-t009:** Results of secondary and compound model tests.

	Quadratic Model	Composite Function Model
Predicted results (10^4^ t)	4035.27	3987.31
Real Results (10^4^ t)	3897.24	3897.24
Prediction error (10^4^ t)	138.03	90.07
Error rate	3.54%	2.31%

**Table 10 ijerph-20-00922-t010:** Gray correlation analysis of carbon emissions and Kaya’s constant decomposition factor.

Factor	Correlation
Population	0.98
Energy use intensity per unit of carbon emissions	0.93
Total energy consumption per 10,000 Yuan GDP	0.89
GDP per capita	0.68

## Data Availability

The data presented in this study are available on request from the corresponding author.

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
