# Peer review of "The Impact of Urban Construction Land Change on Carbon Emissions—A Case Study of Wuhan City"

_ijerph, 2023, doi:10.3390/ijerph20020922_

Round 1

Reviewer 1 Report

The research presented in the paper is interesting and based on solid data and a good review of the literature. I highly appreciate the methodology used in the text and the results.
I perceive the following shortcomings:

1) the introduction lacks a clearly defined purpose of the study;
2) in the discussion, it is necessary to indicate whether the results of the study conducted by the authors are consistent (or not) with the indicated literature;
3) in the conclusion, it is necessary to show what is the originality of the paper in comparison with other studies

Additional comment: reference needed to Xiao Hongyan et al. method (p. 4)

Author Response

Point 1: The introduction lacks a clearly defined purpose of the study.

Response 1: The spatial analysis, curve estimation, correlation evaluation, and grey correlation were used in the article. These methods were used to explore the direct and indirect effects of changes in UCL on carbon emissions, but it is not mentioned in the article, resulting in a lack of clarity in the purpose of the study in the introduction, which we have revised to address this issue.

Point 2: In the discussion, it is necessary to indicate whether the results of the study conducted by the authors are consistent (or not) with the indicated literature;

Response 2: We added a comparison section to the discussion, we compare the results of the study with those of the literature indicated in the introduction and point out the similarities and differences in the research process.

Point 3: In the conclusion, it is necessary to show what is the originality of the paper in comparison with other studies.

Response 3: We have made revisions to address this issue that appeared in the article. We changed the first paragraph of the conclusion to “Based on previous studies, we firstly measures the change in carbon emissions by quantifying the characteristics of urban land use and carbon emissions in Wuhan city; after that, we use spatial autocorrelation analysis and curve estimation, Kaya's constant equation and gray correlation analysis to explore the relationship between spa-tial and temporal changes in UCL on carbon emissions from a spatial perspective; fi-nally, the direct and indirect effects of changes in UCL on carbon emissions are deter-mined.” Responding to the problems of the current study presented in the introduction, and showing the originality of the study in comparison with other studies are presented.

Point 4: Reference needed to Xiao Hongyan et al. method (p. 4)

Response 4: The problems with the references in the article have been corrected.

Reviewer 2 Report

From the Results onward there is a discrepancy in the names of accompanying figures, other then that there are no remarks. 

Author Response

Point 1: From the Results onward there is a discrepancy in the names of accompanying figures, other then that there are no remarks.  

Response 1: The discrepancy in the names of accompanying figures has been corrected. And the necessary elements in the name of the figures and tables are remarked, including the units of the figures and tables contents, time point and time period of the study, etc.

Reviewer 3 Report

This paper explores the direct and indirect effects of urban construction land change on carbon emissions in Wuhan, employing spatial analysis, curve estimation, and correlation evaluation.

I recommend that this paper be accepted after minor revision

1.               The main shortcoming of the article is the absence of a discussion section, where the results of the research are compared with relevant studies (only reference [42] is discussed in section 4). The authors should provide evidence on the question:  "How do the empirical results in this paper differ from those previous findings?" A discussion section is crucial for a complete scientific article.

2.               The authors should add more references and source regarding:

·       The data in Table 2 on Carbon emission factor by energy source

·        On the mathematical model related to spatial correlationn analysis, curve estimation and Kaya's constant equation.

Author Response

Point 1: The main shortcoming of the article is the absence of a discussion section, where the results of the research are compared with relevant studies (only reference [42] is discussed in section 4). The authors should provide evidence on the question:  "How do the empirical results in this paper differ from those previous findings?" A discussion section is crucial for a complete scientific article.

Response 1: We added a comparison section to the discussion, we compare the results of the study with those of the literature indicated in the introduction and point out the similarities and differences in the research process.

Point 2: The authors should add more references and source regarding. (1) The data in Table 2 on Carbon emission factor by energy source. (2) On the mathematical model related to spatial correlationn analysis, curve estimation and Kaya's constant equation.

Response 2: (1) Add the reference to the data in Table 2 on carbon emission factor by energy source. (2) The mathematical model for the spatial correlation analysis analysis has c expressed in Eq. 4 and Eq. 5. The mathematical model related to curve estimation and Kaya's constant equation has been added to Eq. 6, Eq. 7, Eq. 8, Eq. 9, Eq. 10, Eq. 11 and Eq. 12.

Reviewer 4 Report

The study is novel and unique. It helps address a well-defined problem and is structured in professional manner.

I have two comments, and a suggestion

1- I suggest adding a small section to show the reference of studies that used this (or parts) of this methodology  

2- Statistical interpretations must be in the context of the study. Using terms like extremely valid P-value should be avoided. Was multicollinearity considered?  

3- A language editing and review may improve the article 

Author Response

Point 1: I suggest adding a small section to show the reference of studies that used this (or parts) of this methodology.

Response 1: The problems with the references in the article have been corrected. References for these methods, such as The coefficient measurement method, Land use dynamic attitude, Spatial correlation analysis, Curve estimation, Kaya's constant equation and The gray correlation analysis, have been added.

Point 2: Statistical interpretations must be in the context of the study. Using terms like extremely valid P-value should be avoided. Was multicollinearity considered? 

Response 2: (1) R2 and p have been explained in the context of the study. (2) In the study, dependent variable is carbon emissions, and independent variable is UCL area. Since there is only one independent variable, there is no need to consider multicollinearity.

Point 3: A language editing and review may improve the article.

Response 3: This article has been carefully edited by a native English-speaking editor of MogoEdit. Certificate of English editing is in the attachment. 
